# Cyanidin-3-*O*-Galactoside-Enriched *Aronia melanocarpa* Extract Attenuates Weight Gain and Adipogenic Pathways in High-Fat Diet-Induced Obese C57BL/6 Mice

**DOI:** 10.3390/nu11051190

**Published:** 2019-05-27

**Authors:** Su-Min Lim, Hyun Sook Lee, Jae In Jung, So Mi Kim, Nam Young Kim, Tae Su Seo, Jung-Shik Bae, Eun Ji Kim

**Affiliations:** 1Center for Efficacy Assessment and Development of Functional Foods and Drugs, Hallym University, Chuncheon, Gangwon 24252, Korea; sumin8481@hallym.ac.kr (S.-M.L.); wjdwodls79@hallym.ac.kr (J.I.J.); somisss@hallym.ac.kr (S.M.K.); 2Department of Food Science & Nutrition, Dongseo University, Busan 47011, Korea; hyunlee@dongseo.ac.kr; 3R&D center, Wellfine Co., Ltd., Chuncheon, Gangwon 24232, Korea; anold006@wellfine.kr (N.Y.K.); stsu70@wellfine.kr (T.S.S.); bae82045@hanmail.net (J.-S.B.)

**Keywords:** *Aronia melanocarpa*, cyanidin-3-*O*-galactoside, adipogenesis, adipogenic transcription factor, high fat induced obesity

## Abstract

*Aronia melanocarpa* are a rich source of anthocyanins that have received considerable interest for their relations to human health. In this study, the anti-adipogenic effect of cyanidin-3-*O*-galactoside-enriched *Aronia melanocarpa* extract (AM-Ex) and its underlying mechanisms were investigated in an in vivo system. Five-week-old male C57BL/6N mice were randomly divided into five groups for 8-week feeding with a control diet (CD), a high-fat diet (HFD), or a HFD with 50 (AM-Ex 50), 100 (AM-Ex 100), or 200 AM-Ex (AM-Ex 200) mg/kg body weight/day. HFD-fed mice showed a significant increase in body weight compared to the CD group, and AM-Ex dose-dependently inhibited this weight gain. AM-Ex significantly reduced the food intake and the weight of white fat tissue, including epididymal fat, retroperitoneal fat, mesenteric fat, and inguinal fat. Treatment with AM-Ex (50 to 200 mg/kg) reduced serum levels of leptin, insulin, triglyceride, total cholesterol, and low density lipoprotein (LDL)-cholesterol. Real-time reverse transcription-polymerase chain reaction (RT-PCR) analysis revealed that AM-Ex suppressed adipogenesis by decreasing CCAAT/enhancer binding protein α, peroxisome proliferator-activated receptor γ, sterol regulatory element-binding protein-1c, peroxisome proliferator-activated receptor gamma coactivator-1α, acetyl-CoA carboxylase 1, ATP-citrate lyase, fatty acid synthase, and adipocyte protein 2 messenger RNA (mRNA) expressions. These results suggest that AM-Ex is potentially beneficial for the suppression of HFD-induced obesity by modulating multiple pathways associated with adipogenesis and food intake.

## 1. Introduction

The increased prevalence of obesity is a very serious public health problem [1]. Obesity is closely related to the incidence of metabolic syndrome, which includes disorders such as cardiovascular disease, diabetes, hypertension, and hyperlipidemia, and contributes to a lower quality of life and increased mortality [2,3]. Various therapeutic agents for obesity have been developed to improve the condition by controlling appetite, fat absorption, lipid oxidation, the differentiation and proliferation of preadipocytes, lipogenesis, lipolysis, and pro-inflammatory cytokines secretion [4,5,6,7,8]. However, these anti-obesity drugs mostly have side effects or rebound weight gain when the medication is ceased [9,10]. Therefore, there is a great demand for the development of a therapeutic agent for obesity that is effective but has no side effects. 

Polyphenol is a common component in vegetables and fruits, and its variety and abundance are very diverse. Polyphenols exhibit various physiological effects, including antioxidant activity. Since oxidative stress is associated with various diseases, such as insulin resistance, diabetes, cardiovascular disease, and some cancers, plants containing polyphenols are seen as a therapeutic alternative for major diseases in modern people [11,12]. Studies have reported that obesity can be caused by increased oxidative stress and subsequent chronic low-grade inflammatory reactions, and that polyphenol-rich plants can prevent its metabolic complications as well as prevent and treat obesity [13,14,15].

*Aronia melanocarpa*, or chokeberry, is a traditional medicinal herb indigenous to eastern North America and eastern Canada. It is effective for colds and has been used as a hypertension and an arteriosclerosis remedy in Europe and Russia. In Japan, these red-purple fruits have been made into juice, wine, and jam [16]. *Aronia melanocarpa* has various physiological activities, including anti-inflammatory [17], gastroprotective [18], hepatoprotective [19], antidiabetic [20,21], hypolipidemic [22,23], and cardiovascular-protective [24,25] properties. These health improvement effects of *Aronia melanocarpa* are due to the high polyphenol content, which has a high antioxidant effect [16]. The total polyphenols in *Aronia melanocarpa* have been reported to range from 690 to 2560 mg gallic acid equivalent (GAE) per 100 g fresh weight, depending on the extraction conditions as well as cultivation, hydration, and storage conditions of the pulp. Among the anthocyanins of *Aronia melanocarpa* fruit, cyanidin-3- *O* -galactoside is present in the highest amount [16].

By changing the extraction conditions of *Aronia melanocarpa*, we developed an *Aronia melanocarpa* extract with a much higher content of cyanidin-3-*O*-galactoside than that produced by conventional methods. In vitro studies with 3T3-L1 adipocytes have shown that cyanidin-3-*O*-galactoside-enriched *Aronia melanocarpa* extract (AM-Ex) inhibited adipogenesis by down-regulation of adipogenic transcription factors and their target genes [26]. The current study was conducted to confirm the efficacy of AM-Ex in an in vivo system. C57BL/6N mice were used as experimental animals, and dietary obesity was induced by feeding a high fat diet. The effects of AM-Ex in varying concentrations on weight, fat mass, insulin resistance, adipocytokines, blood lipid profiles, and adipogenic transcription factors and their related genes were examined.

## 2. Material and Methods

### 2.1. Preparation of Cyanidin-3-O-Galactoside-Enriched Aronia melanocarpa Extract (AM-Ex)

AM-Ex was prepared by Wellfine Co., Ltd (Hoengseong, Gangwon, Korea). according to the method described previously [26]. In brief, the freeze-dried powder of *Aronia melanocarpa* Nero fruits was extracted with 70% ethanol by adding 100 g of the dried powder to 2 L of 70% ethanol using a high pressure homogenizer (Micronox, Seongnam, Korea) at room temperature. The intermediate extract was additionally extracted under reduced pressure condition using a rotary evaporator in 40 megapascal (Mpa) pressure at 30 °C for 2 h. The extract was concentrated with a rotary evaporator and lyophilized for 72 h in a lyophilizer (IlshinBioBase, Dongducheon, Korea). The resulting powder was used as AM-Ex and stored at −20 °C until further use. The cyanidine-3-*O*-galactoside content of AM-Ex was analyzed using an HPLC (Ultimate 3000, Thermo Fisher Scientific, Waltham, MA, USA). The content of cyanidin-3-*O*-galactoside was 2408 mg /100 g of AM-Ex.

### 2.2. Animals and Treatment

All animal experiments were conducted according to the protocols approved by the Institutional Animal Care and Use Committee of Hallym University (approval number: Hallym 2017-65). Four-week-old male C57BL/6N mice were purchased from Dooyeol Biotech Co. Ltd. (Seoul, Korea) and housed in controlled standard conditions of 23 ± 3 °C temperature, 50 ± 10% relative humidity, and a 12 h light/dark cycle. The mice were acclimated for one week before use and provided free access to a standard non-purified rodent diet (Cargill Agri Purina, Inc., Seongnam, Korea) and water. 

After one week of acclimation, the mice were randomly divided into five groups (10 mice per group): (1) control diet (CD), (2) high-fat diet (HFD), (3) HFD + 50 mg/kg body weight (BW)/day AM-Ex (AM-Ex 50), (4) HFD + 100 mg/kg BW/day AM-Ex (AM-Ex 100), and (5) HFD + 200 mg/kg BW/day AM-Ex (AM-Ex 200). The CD (containing 10 kcal% as fat, No. D12450B) and the HFD (containing 60 kcal% as fat, No. D12452) used in this study were purchased from Research Diets, Inc. (New Brunswick, NJ, USA). The mice were fed for eight weeks and allowed free access to food and water during the entire period. AM-Ex dissolved in physiological saline solution was administered daily by oral gavage for eight weeks. The mice in the CD and the HFD group were given an equal volume of saline solution by oral gavage. Body weights and food intake were measured once a week and daily, respectively, during the feeding period.

At the end of the experimental period, all mice were fasted for 16 h then anesthetized with tribromoethanol diluted with tertiary amyl alcohol. Whole body composition (BW and body fat mass percentage) was determined using dual-energy X-ray absorptiometry (DEXA, PIXImus^TM^, GE Lunar, Madison, WI, USA). Blood was collected from the orbital vein for serum biomarker analysis, after which the adipose tissues were excised, rinsed, and weighed. 

### 2.3. Biochemical Analyses of Blood Samples

Blood was taken from the orbital vein, and serum was obtained by centrifuging the blood at 3000 rpm for 20 min at 4 °C. Serum levels of glucose, triglyceride, total cholesterol, low density lipoprotein (LDL)-cholesterol, and high density lipoprotein (HDL)-cholesterol were measured by a blood chemistry autoanalyzer (KoneLab 20XT, Thermo Fisher Scientific, Vantaa, Finland). Serum insulin levels were measured using an enzyme-linked immunosorbent assay (ELISA) kit (Millipore Corporation, Billerica, MA, USA), and the serum levels of leptin and adiponectin were measured using the relevant ELISA kits (R&D Systems, Minneapolis, MN, USA) according to the manufacturer’s protocols. The homeostasis model assessment of insulin resistance (HOMA-IR) was calculated using the following formula: [fasting glucose (mg/dL) x fasting insulin (mU/L)]/405 [27]. 

### 2.4. Histological Analysis

The epididymal adipose tissues were fixed with 4% paraformaldehyde solution and treated according to a general tissue-processing procedure. The tissue samples were stained with hematoxylin and eosin (H&E) and examined under a light microscope. Photographs were obtained using an AxioImager microscope (Carl Zeiss, Jena, Germany). The size and the number of adipocytes were quantified using an AxioVision Imaging System (Carl Zeiss) with a magnifying power of 200×, and the slides were examined in a blinded manner. 

### 2.5. Quantitative Real-Time Reverse Transcription-Polymerase Chain Reaction

Total RNA from the epididymal adipose tissue was extracted with Trizol (Invitrogen Life Technologies, Carlsbad, CA, USA) according to the manufacturer’s instruction. The content and the purity of the total RNA were estimated using a micro-volume spectrophotometer (BioSpec-nano, Shimadzu, Kyoto, Japan). Quantitative real-time reverse transcription-polymerase chain reaction (RT-PCR) was conducted using a Rotor-gene 3000 PCR (Corbett Research, Mortlake, Australia) and a Rotor-Gene^TM^ SYBR Green kit (Qiagen, Valencia, CA, USA) according to the manufacturer’s instruction. The sequences of the primers used in this study are shown in Table 1. Polymerase chain reaction amplification of the complementary DNA (cDNA) was carried out at 94 °C for 3 min, followed by 40 cycles as follows: 95 °C for 10 s, 60 °C for 15 s, and 72 °C for 20 s. The results were analyzed with Rotor-Gene 6000 Series System Software program, version 6 (Corbett Research, Mortlake, Australia) and normalized to those of glyceraldehyde 3-phosphate dehydrogenase (GAPDH).

### 2.6. Statistical Analyses

All results are presented as the mean ± standard error of the mean (SEM). Statistical analyses were performed using the Student’s t-test to test the differences between the CD group and the HFD group. An ANOVA test followed by Duncan’s multiple comparison test were performed to compare the means between the HFD, the AM-Ex 50, the AM-Ex 100, and the AM-Ex 200 groups. *P* < 0.05 was considered to be statistically significant.

## 3. Results

### 3.1. AM-Ex Reduces Body Weight Gain and Body Fat Mass in HFD-Induced Obese C57BL/6 Mice

The body weights of the mice were measured once a week for eight weeks, as shown in Figure 1. The body weights between the CD and the HFD groups began to differ significantly after the first week of treatment (*P* < 0.001). The body weights in the AM-Ex 200 group were significantly lower than those in the HFD group after the second week of treatment (*P* < 0.05). Meanwhile, the body weights in the AM-Ex 50 and AM-Ex 100 groups were statistically lower than those in the HFD group after the third week of treatment (*P* < 0.05, Figure 1). At the end of the experiment, the HFD group showed dramatically higher body weight gains compared to the CD group (*P* < 0.001). The body weight gains in the three AM-Ex-treated groups were dose-dependently reduced compared to the HFD group (*P* < 0.05). Consistent with the final body weight results, the fat mass percentage by dual energy X-ray absorptiometry (DEXA) analysis revealed that the HFD group showed markedly higher body fat compared to the CD group, and the body fat in the three AM-Ex-treated groups was decreased compared to the HFD group (*P* < 0.05) in a dose-dependent manner (Table 2). There was no significant difference in food intake between the CD and the HFD groups. However, the food efficiency ratio in the HFD group was significantly higher than that in the CD group (*P* < 0.001). The food efficiency ratios in the three AM-Ex-treated groups were significantly lower than in the HFD group (*P* < 0.001) but did not differ significantly among the three AM-Ex-treated groups (Table 2).

### 3.2. AM-Ex Suppresses White Adipose Tissue Accumulation in HFD-Induced Obese C57BL/6 Mice

The white adipose tissue (epididymal, retroperitoneal, mesenteric, and inguinal fat) weights were dramatically increased in the HFD group (*P* < 0.001 for each). In addition, the HFD group showed significantly higher weights of axillary adipose tissues, known as typical brown adipose tissue, compared to the CD group (*P* < 0.001). The AM-Ex treatment significantly decreased the weights of white adipose tissues. However, the axillary adipose tissue weights in the AM-Ex 50, 100, and 200 groups did not differ significantly compared to the HFD group (Figure 2A). We also examined the epididymal adipose tissue histology to determine whether AM-Ex attenuated the hypertrophy of the adipocytes. As shown in Figure 2B, the adipocyte size in the epididymal adipose tissues was larger in the HFD group than in the CD group, and treatment with AM-Ex suppressed enlargement of the adipocytes in the epididymal adipose tissues. The number of adipocytes over 120 μm in the HFD group was markedly higher than in the CD group (*P* < 0.001). The AM-Ex treatment significantly decreased the number of adipocytes over 120 μm in the epididymal adipose tissues (Figure 2C).

### 3.3. AM-Ex Improves Insulin Resistance and Serum Lipid Profiles in HFD-Induced Obese C57BL/6 Mice

Serum glucose levels were markedly increased in the HFD group compared to the CD group (*P* < 0.001). AM-Ex treatment did not affect serum glucose levels, which were elevated by the HFD treatment. Higher insulin levels were observed in the HFD group compared to the CD group. The insulin levels were significantly lowered in the three AM-Ex treated groups. The HOMA-IR index was calculated on the basis of the serum glucose and insulin levels. The HFD group had a dramatically higher HOMA-IR index than the CD group (*P* < 0.01), whereas the AM-Ex treatment markedly decreased the HOMA-IR index at 16.8%, 26.4%, and 38.9% of AM-Ex 50, AM-Ex 100, and AM-Ex 200, respectively, compared to the HFD group (*P* < 0.05). The HFD group had the highest triglyceride, total cholesterol, LDL-cholesterol, and HDL-cholesterol levels. The AM-Ex treatment resulted in a remarkable decrease in triglyceride, total cholesterol, and LDL-cholesterol levels. However, HDL-cholesterol levels did not differ significantly between the HFD group and the three AM-Ex-treated groups (Table 3). 

### 3.4. AM-Ex Reduces Leptin Levels in HFD-Induced Obese C57BL/6 Mice

Compared to the CD group, the HFD group showed higher serum leptin and lower serum adiponectin levels (*P* < 0.001, *P* < 0.01, respectively). The AM-Ex treatment significantly decreased serum leptin levels in a dose-dependent manner (*P* < 0.05). However, there was no significant difference in serum adiponectin levels between the HFD group and the three AM-Ex-treated groups (Figure 3).

### 3.5. AM-Ex Modulates the Expression of Adipogenic Transcription Factors and their Target Genes in the Epididymal Adipose Tissue

To gain insight into the molecular basis of the anti-obesity effect of AM-Ex, we examined the expression levels of adipogenic transcription factors, including CCAAT/enhancer binding protein α (C/EBPα), peroxisome proliferator-activated receptor γ (PPARγ), and sterol regulatory element-binding protein-1c (SREBP-1c) and their target genes, in epididymal adipose tissue. Compared to the CD group, the HFD group exhibited dramatically increased C/EBPα, PPARγ, and SREBP-1c messenger RNA (mRNA) expression (*P* < 0.001). AM-Ex treatment noticeably reduced the mRNA expression of C/EBPα, PPARγ, and SREBP-1c elevated by HFD treatment. In addition, the mRNA expression of peroxisome proliferator-activated receptor gamma coactivator-1α (PGC-1α), a transcriptional coactivator and a key regulator of energy metabolism, was up-regulated in the HFD group compared to the CD group, while it was down-regulated in the three AM-Ex-treated groups (Figure 4). For the genes involved in fatty acid biosynthesis, the mRNA expression of acetyl-CoA carboxylase 1 (ACC1), ATP-citrate lyase (ACL), fatty acid synthase (FAS), and adipocyte protein 2 (aP2) were higher in the HFD group than in the CD group, while they were significantly reduced in the AM-Ex 50, 100, and 200 groups compared to the HFD group. The mRNA expression of carnitine palmitoyltransferase 1 (CPT1), a gene involved in fatty acid oxidation, did not differ significantly between the CD group, the HFD group, and the three AM-Ex-treated groups. There was no obvious difference in mRNA expressions of hormone-sensitive lipase (HSL) and beta-3 adrenergic receptor (β3-AR), which is involved in lipolysis in adipose tissue, between the CD and the HFD groups. However, the mRNA expression of HSL was significantly increased by AM-Ex treatment. AM-Ex treatment increased β3-AR mRNA expression in comparison with the HDF group (6.38-fold, 8.02-fold, and 8.48-fold in the AM-Ex 50, AM-Ex 100, and AM-Ex 200 groups, respectively) (Figure 5).

## 4. Discussion

In recent years, growing attention has been focused on the utilization of natural sources of antioxidants to prevent chronic diseases [12]. *Aronia melanocarpa* represents a lesser-known fruit species utilized mainly in juices, purees, jams, jellies, and wine, as important food colorants, or nutritional supplements [16]. *Aronia melanocarpa*’s antioxidant content is significantly higher than in other berries [28]. The fruit is valued as a great source of antioxidants, especially polyphenols such as phenolic acids (neochlorogenic and chlorogenic acids) and flavonoids (anthocyanins, proanthocyanidins, flavanols, and flavonols) [16,29]. The anthocyanins of *Aronia melanocarpa* are mainly composed of a mixture of four cyanidin glycosides, namely 3-galactoside, 3-glucoside, 3-arabinoside, and 3-xyloside [30], more than 60% of which is cyanidin-3-*O*-galactoside [31]. *Aronia melanocarpa* has a high content of bioactive components and thus demonstrates a number of positive effects, which include strong antioxidant activity and potential medicinal and therapeutic benefits. It also has supportive impacts on lipid profiles, fasting plasma glucose, and blood pressure levels. Therefore, it could also be useful in the prevention and the treatment of chronic diseases, such as metabolic disorders, diabetes, and cardiovascular diseases.

The anthocyanin content of *Aronia melanocarpa* is affected by genetic factors (cultivar, genotype, variety) and growth conditions (location, cultivation technique, ripening stage), processing, and storage [16]. The concentration of anthocyanin in *Aronia melanocarpa* extract also varies greatly depending on the extraction method. To increase the antioxidant capacity of *Aronia melanocarpa*, a method that extracts the maximum amount of anthocyanin is desirable. As previously reported, we obtained *Aronia melanocarpa* extracts with high cyanidin-3-*O*-galactoside content (AM-Ex) by adding pressure condition manipulations to existing extraction methods. When extracted with the new extraction method, the content of cyanidin-3-*O*-galactoside was 2408 mg/100 g, which was about 2.4 times higher than that of conventional methods. Our in vitro studies using 3T3-L1 cells showed that this AM-Ex inhibited adipogenesis [26]. This study was conducted to investigate the anti-obesity properties of cyanidin-3-*O*-galactoside-enriched AM-Ex in an in vivo system.

High-fat diet-fed mice are used as an animal model to study metabolic syndrome, including obesity, because it increases abdominal fat, blood lipids, liver enzyme activity, liver weight, the inflammatory response, and collagen deposition [32]. We examined the effect of AM-Ex supplementation of an 8-week high-fat diet on body weight and body fat mass in C57BL/6 mice. As a result, the increase in body weight and fat mass due to high-fat diet consumption was dose-dependently inhibited by AM-Ex supplementation (Figure 1). As shown in Figure 2, the weight of white fat tissue (epididymal fat, retroperitoneal fat, mesenteric fat, and inguinal fat) was dose-dependently decreased by AM-Ex supplementation in C57BL/6 mice. This is similar to other studies that reported that supplementation with Aronia fruit [33] and cyanidin glycoside [34] in high-fat-fed animals reduced body weight gain, visceral adiposity index, and total body fat mass. In addition, a human study reported that Aronia juice intake was effective in improving obesity [21,35]. In the current study, unlike white adipose tissue, axillary fat, a brown adipose tissue, was significantly increased in mice fed the HFD compared to the CD (*P* < 0.001), but supplementation with varying concentrations of AM-Ex did not affect the axillary fat mass (Figure 2). Regarding the adipocyte size of the epididymal fat pad, hypertrophy of the adipocyte was significantly increased in the HFD group compared to the CD group, which was suppressed by AM-Ex supplementation (Figure 2B,C). Therefore, the decrease in body fat and weight due to supplementation with AM-Ex can be attributed to the decrease of white adipose tissue and the suppression of adipocyte hypertrophy.

It has already been reported that anthocyanins of *Aronia melanocarpa* decreased the absorption of sugars and lipids in the digestive system, thereby preventing obesity and improving diabetes. *Aronia melanocarpa* inhibits the development of diabetes by preventing the postprandial hyperglycemia by inhibiting the activity of α-glucosidase and α-amylase [21]. Among the anthocyanins of *Aronia melanocarpa*, cyanidin-3-*O*-rutinoside inhibits the actions of pancreatic α-amylase, lipase, and α-glucosidase and thus improves glucose metabolism and prevents diabetes [36]. In our study, blood glucose levels were significantly increased in the HFD group compared to the CD group (*P* < 0.001). However, supplementation with 50-200 mg/kg BW of AM-Ex did not reduce blood glucose levels in HFD fed mice (Table 3). Our results differ from other studies that reported that *Aronia melanocarpa* extract significantly reduced blood glucose levels [21,37]. The subjects used in these studies were diabetic or pre-diabetic animal models. Because we used non-diabetic experimental animals that maintained normal blood sugar levels, this may have caused the difference in results from previous studies. In our study, blood insulin levels were decreased, and insulin resistance was significantly improved in proportion to the AM-Ex concentration (Table 3). AM-Ex appeared to reduce the risk factors associated with insulin resistance by controlling multiple pathways associated with insulin signaling, adipogenesis, and inflammation. Insulin sensitivity has been demonstrated in both human and animal studies to be directly linked to weight loss [38,39]. According to Vlachojannis et al. [40], the minimum level of anthocyanin required to improve the metabolic syndrome is 110 mg per day. *Aronia melanocarpa* has also been shown to improve insulin resistance and improve lipid metabolism [23,35,41]. Kim et al. [41] reported that *Aronia melanocarpa* extract decreased FAS and ACC gene expression associated with lipid and lipoprotein metabolism, as well as the cholesterol synthesis-linked gene expression of SREBP, scavenger receptor class B1, and ATP-binding cassette transporter A1. In a study of subjects with mild hypercholesterolemia, Skoczyriska et al. [42] found that the use of Aronia juice reduced blood LDL-cholesterol and increased HDL-cholesterol. In the current study, AM-Ex treatment was also effective in reducing serum triglyceride, total cholesterol, and LDL-cholesterol levels (Table 3). However, unlike the results of Skoczyriska et al. [42], our study did not show an increase in HDL-cholesterol (Table 3). 

Adipogenesis is regulated by transcription factors such as C/EBPs, PPAR-γ, and SREBP-1c, and transcription coactivators, such as PGC-1α. They act as key regulators of glucose and energy metabolism as well as regulation of preadipocyte differentiation and lipid biosynthesis. PGC-1α interacts with the nuclear receptor PPAR-γ, which permits the interaction of this protein with multiple transcription factors [43,44,45,46,47]. In our study, HFD-fed mice showed significant increases in C/EBPα, SREBP-1c, PPAR-γ, and PGC-1α compared to CD (*P* < 0.001). Supplementation of HFD with AM-Ex decreased both transcription and transcription coactivators. C/EBPs was significantly reduced by 50 mg/kg BW AM-Ex supplementation, and the effect remained the same for supplementation amounts of 100 and 200 mg/kg BW AM-Ex. SREBP-1c, PPAR-γ, and PGC-1α were not significantly different at 50 mg/kg BW AM-Ex treatment, and mRNA expression was significantly decreased at 100 and 200 mg/kg BW AM-Ex supplementation (Figure 4). Specific adipogenesis-related genes affected by these transcription factors, such as ACC1, ACL, FAS, and aP2, were also down-regulated by AM-Ex supplementation (Figure 5). aP2 is used as an early biomarker of metabolic syndrome. Increased gene expression of aP2 is associated with an increase in cardiovascular disease, diabetes, and obesity [48,49,50]. In this study, aP2 increased by HFD ingestion was significantly decreased by AM-Ex supplementation, and 200 mg/kg BW AM-Ex lowered it to the level of the CD group (Figure 5). ACC promotes the formation of malonyl-CoA to induce new fatty acid synthesis [51]. ACL is an enzyme that catalyzes the hydrolysis of ATP and is involved in the synthesis of triglyceride and cholesterol [52]. FAS is activated as a key enzyme in de novo lipogenesis, and when activated, it increases insulin resistance as well as fatty acid synthesis in adipose tissues [43]. Qin and Anderson [53] reported that chokeberry extracts up-regulated PPAR-γ and adiponectin expression and down-regulated aP2 and FAS in Wister rats fed a high fructose diet. Guo et al. [54] demonstrated that cyanidin-3-*O*-glucoside activated the AMP-activated protein kinase-dependent signaling pathway in human HepG2 cells to promote ACC phosphorylation, thereby increasing fatty acid oxidation and improving glucose metabolism. To date, the major potential anti-adipogenic bioactive molecules recognized are oxysterol, (−)-epigallocatechin, genistein, and resveratrol [55]. Based on previous studies and our study, the cyanidin-3-*O*-glycosides are also worth including here. 

Adipose tissue is a dynamic organ that secretes adipokines. Adipokines play a crucial role in regulating insulin sensitivity, adipogenesis, and obesity. The dysregulated secretion of adipokines results in obesity and insulin resistance. Leptin is as an obese gene product in control of energy balance by inhibiting hunger—generally in proportion to fat mass—with rises in serum levels resulting in a decrease in food intake and an increase in energy expenditure [56,57]. In our study, serum leptin levels in the HFD group were increased compared to the CD group, and AM-Ex treatment significantly reduced serum leptin levels in a dose-dependent manner (Figure 3). Similar results were reported for chokeberry extract [53] and cyanidin-3-*O*-glucoside [58]. In contrast, serum adiponectin concentrations were significantly reduced in the HFD group compared to the CD group, but supplementation with AM-Ex did not normalize this decrease (Figure 3). As the leptin concentration increases, appetite and adipogenesis both increase, whereas the increase in adiponectin concentration increases insulin sensitivity and lipolysis, thereby improving obesity [59,60]. In addition, in our study, food intake and food efficiency decreased in proportion to the concentration of AM-Ex treatment in C57BL/6 mice (Table 2). It can be suggested that changes in the concentrations of adipokines due to Am-Ex treatment were partially related to this mechanism. Additionally, AM-Ex seemed to have more effect on leptin than adiponectin, and the decrease in adipose tissue weight and blood profile improvement by AM-Ex supplementation may have been related to decreases in serum leptin concentrations. 

In our study, adipogenesis-related transcription factors and coactivators and their associated gene expression were significantly reduced by AM-Ex supplementation, whereas the genes associated with lipolysis were only partially altered (Figure 5). We examined the gene expression of CPT-1 involved in fatty acid oxidation and that of HSL and β3-AR involved in lipolysis in adipose tissue. The results showed that HFD did not significantly change these three genes compared to CD. In the case of CPT-1, there was no change, regardless of the supplementation amount from 50 to 200 mg/kg BW of AM-Ex. The gene expressions of HSL and β3-AR were increased but were not dose-dependent, since the values were nearly the same or more than those in the group supplemented with 50 mg/kg BW Am-Ex. These results suggest that the effect of AM-Ex supplementation on fat mass reduction may be due to the inhibition of adipogenesis rather than the stimulation of lipolysis. 

## 5. Conclusions

To summarize the results of this study, we found that cyanidine-3-*O*-galactoside-enriched AM-Ex had an anti-obesity effect in C57BL/6 mice. AM-Ex supplementation reduced insulin resistance, serum triglyceride, LDL- and total cholesterol levels, white fat tissue weights, and leptin levels. These changes were related to adipogenesis-related transcription factors and coactivators and the down-regulation of specific adipogenesis-related genes affected by these transcription factors. These changes were also related to decreased food intake due to changes in leptin levels. Our results suggest that cyanidine-3-*O*-galactoside-enriched *Aronia melanocarpa* may be used as an anti-obesity agent. More research is needed to find effective and safe doses to apply these results to humans in the future.

## Figures and Tables

**Figure 1 nutrients-11-01190-f001:**
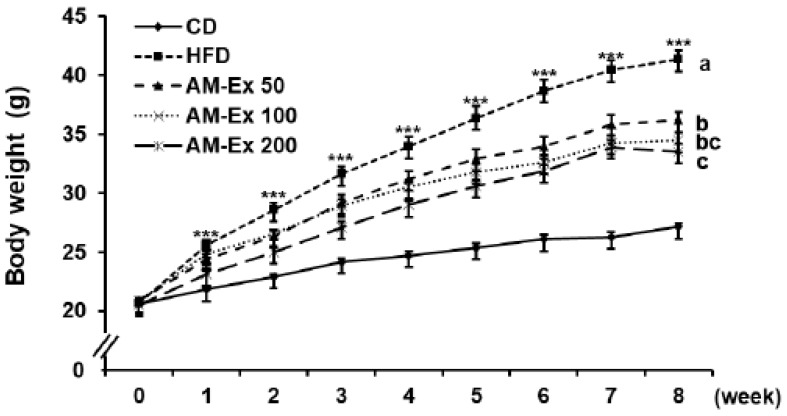
Effect of *Aronia melanocarpa* extract (AM-Ex) administration on body weight in high fat diet (HFD)-fed C57BL/6N mice. AM-Ex was administered by oral gavage for eight weeks to mice fed the HFD. Body weights were measured every week. Each body weight point represents the mean ± standard error of the mean (SEM) (n = 10). *** *P* < 0.001 significantly different from the control diet (CD) group. Means without a common letter differ between HFD, AM-Ex 50, AM-Ex-100, and AM-Ex 200 groups at *P* < 0.05.

**Figure 2 nutrients-11-01190-f002:**
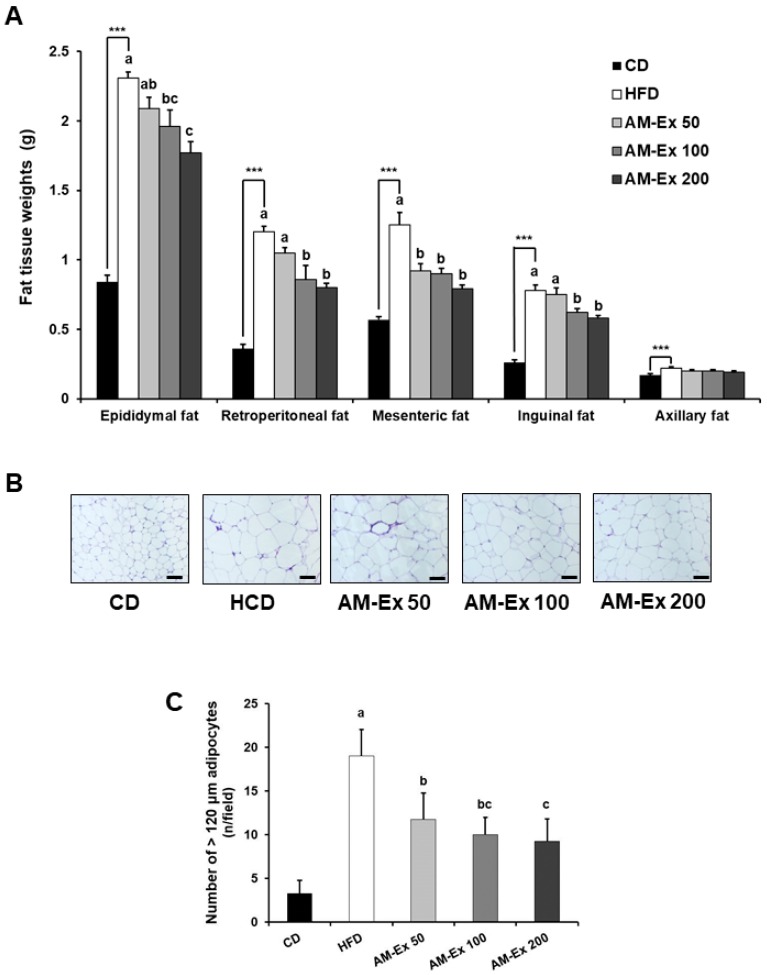
Effect of AM-Ex administration on weight of various adipose tissues and morphological changes in epididymal adipose tissue in HFD-fed C57BL/6N mice. AM-Ex was administered by oral gavage for eight weeks to mice fed HFD. (**A**) Adipose tissue weights in epididymal, retroperitoneal, mesenteric, inguinal, and axillary fat. Each bar represents the mean ± SEM (n = 10). (**B**) Representative hematoxylin and eosin (H&E) stained images of epididymal adipose tissues (n = 10). (**C**) The number of adipocytes over 120 μm was counted. Each bar represents the mean ± SEM (n = 10). 200 x magnification, Scale bar = 100 μm. *** *P* < 0.001 significantly different from the CD group. Means without a common letter differ between HFD, AM-Ex 50, AM-Ex 100, and AM-Ex 200 groups at *P* < 0.05.

**Figure 3 nutrients-11-01190-f003:**
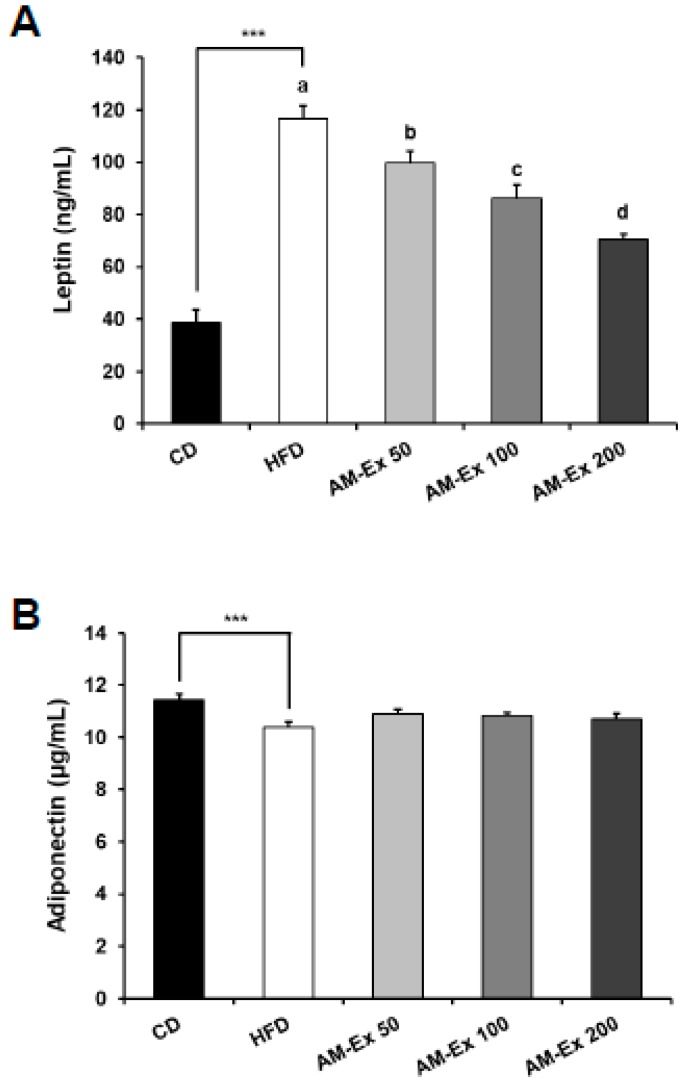
Effect of AM-Ex administration on the serum leptin and adiponectin levels in HFD-fed C57BL/6N mice. AM-Ex was administered by oral gavage for eight weeks while the mice were fed the HFD. The serum levels of leptin (**A**) and adiponectin (**B**) were measured with the appropriate enzyme-linked immunosorbent assay (ELISA) kit. Each bar represents the mean ± SEM (n = 10). *** *P* < 0.001 significantly different from the CD group. Means without a common letter differ between HFD, AM-Ex 50, AM-Ex 100, and AM-Ex 200 groups at *P* < 0.05.

**Figure 4 nutrients-11-01190-f004:**
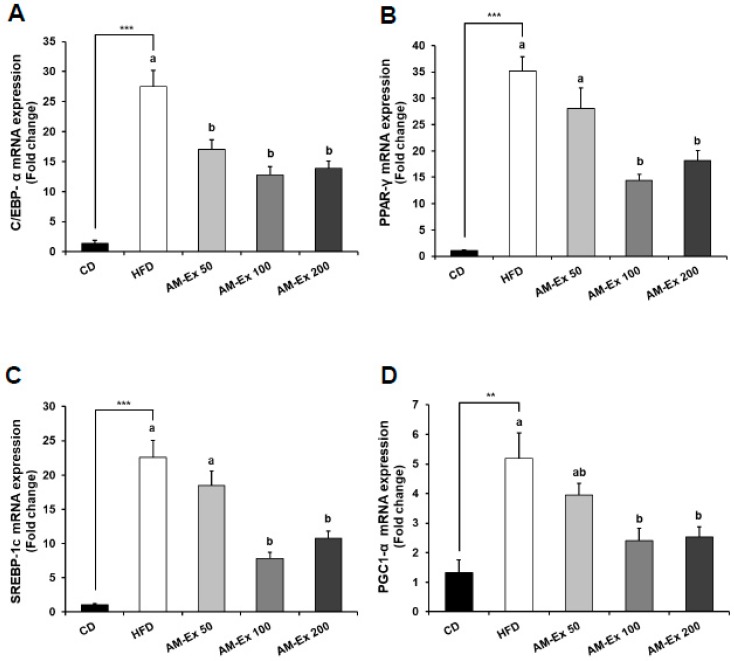
Effect of AM-Ex administration on the expression of adipogenic transcription factors in epididymal adipose tissue of HFD-fed C57BL/6N mice. AM-Ex was administered by oral gavage for eight weeks while the mice were fed the HFD. The total RNA in epididymal adipose tissue was isolated, reverse transcribed, and real-time PCR was conducted. The amount of each messenger RNA (mRNA) was normalized to the amount of glyceraldehyde 3-phosphate dehydrogenase (GAPDH) mRNA and expressed relative to the CD group. Each bar represents the mean ± SEM (n = 10). ** *P* < 0.01, *** *P* < 0.001 significantly different from the CD group. Means without a common letter differ between HFD, AM-Ex 50, AM-Ex 100, and AM-Ex 200 groups at *P* < 0.05.

**Figure 5 nutrients-11-01190-f005:**
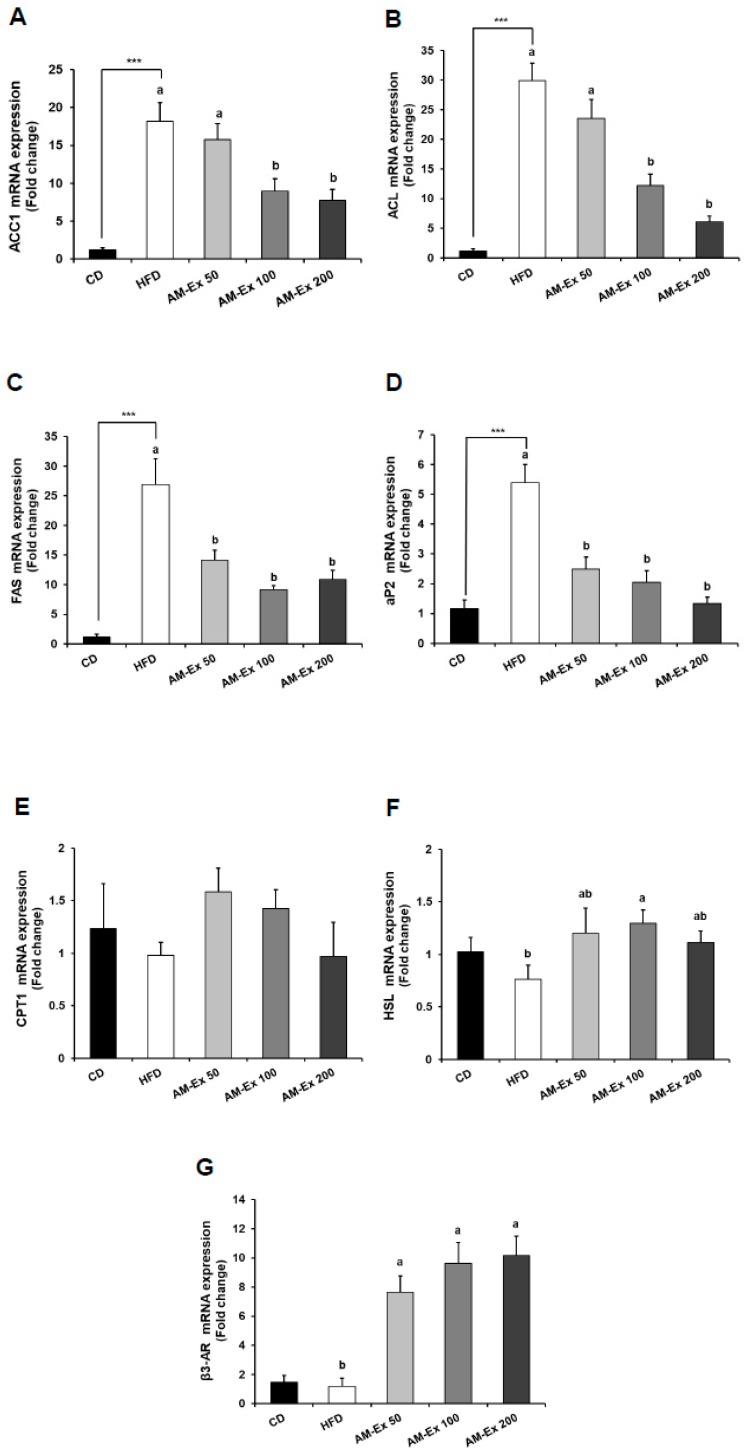
Effect of AM-Ex administration on the expression of adipogenesis-related genes in epididymal adipose tissue of HFD-fed C57BL/6N mice. AM-Ex was administered by oral gavage for eight weeks while the mice were fed the HFD. The total RNA in epididymal adipose tissue was isolated, reverse transcribed, and real-time PCR was conducted. The amount of each mRNA was normalized to the amount of GAPDH mRNA and expressed relative to the CD group. Each bar represents the mean ± SEM (n = 10). *** *P* < 0.001 significantly different from the CD group. Means without a common letter differ between HFD, AM-Ex 50, AM-Ex 100, and AM-Ex 200 groups at *P* < 0.05.

**Table 1 nutrients-11-01190-t001:** Primer sequences used in this study.

Target Gene	Forward Primer (5′-3′)	Reverse Primer (5′-3′)
ACC1	GGAGATGTACGCTGACCGAGAA	ACCCGACGCATGGTTTTCA
ACL	TGGATGCCACAGCTGACTAC	GGTTCAGCAAGGTCAGCTTC
aP2	GGATTTGGTCACCATCCGGT	TTCACCTTCCTGTCGTCTGC
β3-AR	CCTAGCTGTCACCAACCCTTT	GACGAAGAGCATCACAAGGAG
C/EBPα	TGGACAAGAACAGCAACGAGTAC	TGGACAAGAACAGCAACGAGTAC
CPT1	GTGCTGGAGGTGGCTTTGGT	TGCTTGACGGATGTGGTTCC
FAS	AGGGGTCGACCTGGTCCTCA	GCCATGCCCAGAGGGTGGTT
HSL	CCGTTCCTGCAGACTCTCTC	CCACGCAACTCTGGGTCTAT
PGC-1α	CCCTGCCATTGTTAAGACC	TGCTGCTGTTCCTGTTTTC
PPARγ	CAAAACACCAGTGTGAATTA	ACCATGGTAATTTCTTGTGA
SREBP-1c	CACTTCTGGAGACATCGCAAAC	ATGGTAGACAACAGCCGCATC
GAPDH	AGGTTGTCTCCTGCGACT	TGCTGTAGCCGTATTCATTGTCA

**Table 2 nutrients-11-01190-t002:** Effect of AM-Ex on body weight gain, fat mass percentage, food intake, and food efficiency ratio in HFD-fed C57BL/6N mice.

	Groups	CD	HFD	AM-Ex 50	AM-Ex 100	AM-Ex 200
Variables						
Initial body weight (g)		20.5 ± 0.2	20.6 ± 0.4	20.9 ± 0.2	20.9 ± 0.1	20.5 ± 0.3
Final body weight (g)		27.1 ± 0.3	41.3 ± 0.8 ***^,a^	36.2 ± 0.7 ^b^	34.4 ± 0.7 ^bc^	33.5 ± 0.6 ^c^
Body weight gain (g)		6.6 ± 0.4	20.7 ± 0.9 ***^,a^	15.3 ± 0.7 ^b^	13.6 ± 0.7 ^bc^	13.0 ± 0.6 ^c^
Fat mass percentage (%)		27.8 ± 0.6	44.2 ± 0.6 ***^,a^	41.5 ± 0.6 ^b^	39.1 ± 0.8 ^c^	36.9 ± 0.6 ^d^
Food intake (g/day)		2.32 ± 0.01	2.30 ± 0.02 ^a^	2.06 ± 0.03 ^b^	2.00 ± 0.02 ^bc^	1.96 ± 0.02 ^c^
Food efficiency ratio ^1)^		0.051 ± 0.004	0.161 ± 0.007 ***^,a^	0.133 ± 0.005 ^b^	0.121 ± 0.005 ^b^	0.118 ± 0.005 ^b^

^1)^ Food efficiency ratio = weight gain/food intake. The values are expressed as mean ± standard error of the mean (SEM) (n = 10). *** *P* < 0.001 significantly different from the control diet (CD) group. Means without a common letter differ between high fat diet (HFD, *Aronia melanocarpa* extract (AM-Ex 50, AM-Ex 100, and AM-Ex 200 groups at *P* < 0.05.

**Table 3 nutrients-11-01190-t003:** Effect of AM-Ex on insulin resistance and serum lipid levels in HFD-fed C57BL/6N mice.

	Groups	CD	HFD	AM-Ex 50	AM-Ex 100	AM-Ex 200
Variables						
Glucose (mg/dL)		163.6 ± 6.7	201.6 ± 3.8 ***	206.6 ± 4.6	190.8 ± 4.3	189.4 ± 8.7
Insulin (ng/mL)		4.16 ± 0.90	9.21 ± 1.49 **^,a^	7.47 ± 0.78 ^ab^	7.12 ± 0.47 ^ab^	5.99 ± 0.51 ^b^
HOMA-IR ^1)^		39.4 ± 8.3	109.2 ± 17.4 **^,a^	90.8 ± 9.3 ^ab^	80.4 ± 5.4 ^ab^	66.7 ± 5.9 ^b^
Triglyceride (mg/dL)		51.8 ± 3.0	71.6 ± 4.6 **^,a^	62.9 ± 4.4 ^ab^	55.2 ± 4.1 ^bc^	44.5 ± 1.5 ^c^
Total cholesterol (mg/dL)		129.6 ± 4.7	170.6 ± 5.5 ***^,a^	153.4 ± 3.9 ^bc^	161.8 ± 5.3 ^ab^	147.1 ± 3.9 ^c^
LDL-cholesterol (mg/dL)		23.7 ± 1.9	38.8 ± 3.7 ***^,a^	30.1 ± 2.1 ^b^	32.1 ± 1.2 ^ab^	31.5 ± 1.5 ^b^
HDL-cholesterol (mg/dL)		90.1 ± 3.4	119.6 ± 6.0 ***	113.5 ± 3.1	123.7 ± 3.4	111.9 ± 4.1

^1)^ Homeostasis model assessment of insulin resistance (HOMA-IR) was calculated on the basis of the formula: [fasting glucose (mg/dL) x fasting insulin (mU/L)]/405. The values are expressed as mean ± SEM (n = 10). ** *P* < 0.01, *** *P* < 0.001 significantly different from the CD group. Means without a common letter differ between HFD, AM-Ex 50, AM-Ex 100, and AM-Ex 200 groups at *P* < 0.05. LDL: Low density lipoprotein; HDL; High density lipoprotein.

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
