# Peer review of "Cyanidin-3-O-Galactoside-Enriched Aronia melanocarpa Extract Attenuates Weight Gain and Adipogenic Pathways in High-Fat Diet-Induced Obese C57BL/6 Mice"

_nutrients, 2019, doi:10.3390/nu11051190_

Round 1

Reviewer 1 Report

The manuscript by Lim et al reported the effects of Aronia Melanocarpa on mice fed high fat diet.

Additional comments are appended below :

- Conclusions are mostly based on the change of expressions of genes, which is not enough. It would be better to investigate protein expression by WB or ELISA in addition to the mRNA panel. 

- Discussion of the results must be improved.

- Information about PGC-1a is lacked in this manuscript.

- Line 151: Specify which table contains dexa analysis.

- In figure 2 B and C, the adipocyte number and adipocyte area should be clarified. Data on adipocytes area is missing and should support figure 2B.

- Figure 5: HSL mRNA levels seem similar between the different groups. The scale is very small.

- Authors should measure more markers related to beta oxidation such as AMPK or PPAR alpha.

- The manuscript is descriptive and no mechanisms are associated to the effects of AM-EX

Author Response

Response Letter

Dear

We have pleased to resubmit the revision of our manuscript “Cyanidin-3-O-galactoside-enriched Aronia melanocarpa extract attenuates weight gain and adipogenic pathways in high-fat diet-induced obese C57BL/6 mice”. We would like to appreciate your and the expert’s valuable and thoughtful comments. Therefore, we carefully revised our manuscript. The detailed responses, we made, to each of your comments are provided below. In the revised manuscript, corrected parts are written in red colors.

Sincerely,

Eun Ji Kim, Ph.D.

Research Professor

Center for Efficacy Assessment and Development of Functional Foods and Drugs

Hallym University, Chuncheon, 24252 Republic of Korea

Tel: +82-33-248-3106

Fax: +82-33-244-3107

----------------------------------------------------------------------------------------------------------------

Response to Reviewer 1 Comments

Point 1:  Conclusions are mostly based on the change of expressions of genes, which is not enough. It would be better to investigate protein expression by WB or ELISA in addition to the mRNA panel.

Response 1: We totally agree with this opinion. However, protein expression in adipose tissues was not available within the revision period (within 10 days) because it takes a long time to analyse proteins. There are many cases in which only mRNA expression of genes associated with obesity is examined as a basis for suggesting the anti-obesity mechanism in many papers (Food and Chemical Toxicology 125: 85-94, 2019; Food and Chemical Toxicology 106: 393-403, 2017; Journal of Nutritional Biochemistry 23: 1732-1739, 2012; Obesity 17: 2127-2133, 2009). Thus, although it is not perfect, mRNA expression of the gene may explain the anti-obesity mechanism of AM-Ex. We would like the reviewer to understand the point.

Point 2: Discussion of the results must be improved.

Response 2: We have improved the discussion.

Point 3: Information about PGC-1a is lacked in this manuscript.

Response 3: We have supplemented the function of PGC-1a as transcription coactivators in our manuscript.

Point 4: Line 151: Specify which table contains DEXA analysis.

Response 4: We have specified which table contains DEXA analysis.

Consistent with the final body weight results, the fat mass percentage by DEXA analysis revealed that the HFD group showed markedly higher body fat compared to the CD group and the body fat in the three AM-Ex-treated groups was decreased compared to the HFD group (P < 0.05) in a dose-dependent manner (Table 2).

Point 5: In Figure 2B and C, the adipocyte number and adipocyte area should be clarified. Data on adipocytes area is missing and should support Figure 2B.

Response 5: To clarify the adipocyte number and area in Figure 2B and C, we have added a scale bar (100 mm) in H&E stained imagines and revised the legends of the Y-axis in Figure C to ‘Number of  >  120 mm adipocytes (n/field)’

Point 6: Figure 5. HSL mRNA levels seem similar between the different groups. The scale is very small.

Response 6: Compared to other genes, the expression of HSL mRNA expression was lower. However, the analysis of statistical significance revealed differences between the different experimental groups.

Point 7: Authors should measure more markers related to beta oxidation such as AMPK or PPAR a.

Response 7: We investigated the expression of various obesity-related genes to elucidate the anti-obesity mechanisms of AM-Ex. We have determined in this paper that among the various genes, the genes that has been significantly altered by AM-Ex treatment are critical genes for AM-Ex’s anti-obesity effect. For example, AM-Ex did not affect the mRNA expression of PPARa, LPL, AMPK, UCP1 and UCP3.

Point 8: The manuscript is descriptive and no mechanisms are associated to the effects of AM-Ex.

Response 8: The results of this study suggest that the mechanism of body fat mass reduction by AM-Ex appears to be due to decreased adipogenesis and food intake, and this is supplemented in the discussion.

Reviewer 2 Report

Lim et al investigated the effects of Cyanidin-3-O-galactoside-enriched Aronia melanocarpa extract (AM-Ex) on HF diet-induced obesity. It has a simple study deign demonstrating that 50-200 mg/Kg BW of AM-Ex attenuates the adiposity presumably due to reduced adipogenesis.  Most results are quite straight forward. Below are several points that require clarification.

1.      Fasting glucose and fasting insulin levels are outrageously high compared with other HF-feeding studies. As a consequence, HOMA-IR values become unusually high. Please check the values and unit to calculate the right range of HOMA-IR.

2.      There was a significant decrease of food intake but it was not carefully addressed.

3.      HSL is also one of target genes of PPARg. It is little confusing why HSL expression levels are particularly lower than other PPARg target genes.

4.      Decreased adipose tissue mass is usually due to either 1) increased fatty acid oxidation and energy expenditure, 2) decreased intake and 3) increased hepatic lipid accumulation. Discussion should be handled these different fate of fats upon AM-Ex supplementation.

5.      Epididymal fat is not usually active for adipose tissue browning. The potential possibility of adipose tissue browning could be investigated by using the inguinal adipose tissue or thermogenic potential of brown adipose tissue.

Author Response

Response Letter

Dear

We have pleased to resubmit the revision of our manuscript “Cyanidin-3-O-galactoside-enriched Aronia melanocarpa extract attenuates weight gain and adipogenic pathways in high-fat diet-induced obese C57BL/6 mice”. We would like to appreciate your and the expert’s valuable and thoughtful comments. Therefore, we carefully revised our manuscript. The detailed responses, we made, to each of your comments are provided below. In the revised manuscript, corrected parts are written in red colors.

Sincerely,

Eun Ji Kim, Ph.D.

Research Professor

Center for Efficacy Assessment and Development of Functional Foods and Drugs

Hallym University, Chuncheon, 24252 Republic of Korea

Tel: +82-33-248-3106

Fax: +82-33-244-3107

----------------------------------------------------------------------------------------------------------------

Response to Reviewer 2 Comments

Point 1:  Fasting glucose and fasting insulin levels are outrageously high compared with other HF-feeding studies. As a consequence, HOMA-IR values become unusually high. Please check the values and unit to calculate the right range of HOMA-IR.

Response 1: We knew that fasting glucose, fasting insulin, and HOMA-IR are high compared with other HF-feeding studies. So we have checked the values and units and confirmed that nothing was wrong. Several papers reported that fasting glucose, fasting insulin, and HOMA-IR differed according to various circumstances, including animal species and diet. Several papers (Journal of Ethnopharmacology 183: 95-102, 2016; PLOS ONE 10 (1): e0117556, 2015) reported fasting glucose, fasting insulin, and HOMA-IR values similar to our paper. This indicates that our results are not wrong.

Point 2:  There was a significant decrease of food intake but it was not carefully addressed.

Response 2: Body weight reduction by AM-Exis associated with a decrease in adipogenesis as well as a decrease in food intake, which may be related to serum leptin levels. We have supplemented this in discussion.

Point 3:  HSL is also one of target genes of PPARg. It is little confusing why HSL expression levels are particularly lower than other PPARg target genes.

Response 3: This appears to be due to AM-Ex affecting adipogenesis-related gene expression rather than lipolysis, as presented in discussion (line 388-398).

Point 4:  Decreased adipose tissue mass is usually due to either 1) increased fatty acid oxidation and energy expenditure, 2) decreased intake and 3) increased hepatic lipid accumulation. Discussion should be handled these different fate of fats upon AM-Ex supplementation.

Response 4: Thank you for your good point. AM-Ex appears to reduce body fat mass through reduced food intake and adipogenesis. Manuscript has been modified in this direction.

Point 5:  Epididymal fat is not usually active for adipose tissue browning. The potential possibility of adipose tissue browning could be investigated by using the inguinal adipose tissue or thermogenic potential of browning adipose tissue. 

Response 5: In this study, we measured the weight of axillary adipose tissue to determine whether AM-Ex affects adipose tissue browning, and found that AM-Ex did not affect the weight of axillary adipose tissue (adipose tissue browning). In addition, we found no change in adipose tissue browning in the inguinal adipose tissue. In order to study anti-obesity mechanisms of AM-Ex, except for the possibility of adipose tissue browning, we determined mRNA expression changes in epididymal

Round 2

Reviewer 1 Report

Authors replied to majority of reviewer's comments

Reviewer 2 Report

No more specific concerns